# ACCELERATING INVERSE REINFORCEMENT LEARNING WITH EXPERT BOOTSTRAPPING

## ABSTRACT

Existing inverse reinforcement learning methods (e.g. MaxEntIRL, $f$-IRL) search over candidate reward functions and solve a reinforcement learning problem in the inner loop. This creates a rather strange inversion where a harder problem, reinforcement learning, is in the inner loop of a presumably easier problem, imitation learning. In this work, we show that better utilization of expert demonstrations can reduce the need for hard exploration in the inner RL loop, hence accelerating learning. Specifically, we propose two simple recipes: (1) placing expert transitions into the replay buffer of the inner RL algorithm (e.g. Soft-Actor Critic) which directly informs the learner about high reward states instead of forcing the learner to discover them through extensive exploration, and (2) using expert actions in Q value bootstrapping in order to improve the target Q value estimates and more accurately describe high value expert states. Our methods show significant gains over a MaxEntIRL baseline on the benchmark MuJoCo suite of tasks, speeding up recovery to 70% of deterministic expert performance by 2.13x on HalfCheetah-v2, 2.6x on Ant-v2, 18x on Hopper-v2, and 3.36x on Walker2d-v2.

## 1 INTRODUCTION

The core problem in inverse reinforcement learning (IRL) is to recover a reward function that explains the expert's actions as being optimal, and a policy that is optimal with respect to this reward function, thus matching expert behavior. Existing methods like MaxEntIRL (Ziebart et al., 2008) and $f$-IRL (Ni et al., 2020) accomplish this by running an outer-loop that updates a reward function and an inner-loop that runs reinforcement learning (RL), usually many steps of policy iteration.

However, running RL in the inner-loop results in high sample and computational complexity compared to IL (Sun et al., 2017). Specifically, this requires large numbers of learner rollouts. Learner rollouts can be expensive, especially with high fidelity, complex simulators, and in the real world, where excessive rollouts can lead to an elevated risk of damage to the physical agent or system. It is therefore important to study methods for accelerating the inner RL loop. Our key insight is that *instead of treating the inner RL as some black box policy optimization, we can provide valuable information about potentially high reward regions that can significantly accelerate learning*.

We propose two simple recipes that are applicable to a wide class of inner RL solvers, notably any actor-critic approaches (e.g. Soft-Actor Critic (SAC) (Haarnoja et al., 2018)):

1. Place expert transitions into the actor's replay buffer. These transitions contain high reward states that accelerate policy learning and reduce the amount of exploration required to discover such high reward states. We call this method *expert replay bootstrapping (ERB)*.

2. Use the expert's *next action* from each transition in the critic's target Q-value estimator. By leveraging such side information, we more accurately describe high value expert states and improve the estimate of the next state's target value. We call this method *expert Q bootstrapping (EQB)*.

In general, the critic's target value estimate is derived from the actor and the actor's optimization objective is derived from the critic. This creates a mutual bond where neither side can move forward without the other progressing accordingly. When the policy is lagging behind due to its inability to effectively maximize a potentially complex Q function surface, the policy's action may be quite

suboptimal. This creates lower critic target value estimates for expert states and slows learning. However, in the imitation learning setting, we have access to expert demonstrations that allow the critic to progress in its learning without being held back by the actor. In fact, the critic's value function surface is exactly derived from these expert demonstrations - hence, the expert demonstrations are maximizing value. By leveraging expert Q bootstrapping and providing accurate targets using the expert's next action, we allow the critic to progress as if it were working with a stronger policy. *Accurate targets allow the Q function to progress in its learning and provide better signals to the policy, further accelerating learning.*

We show that our methods are able to accelerate multiple state-of-the-art inverse reinforcement learning algorithms such as MaxEntIRL (Ziebart et al., 2008) and $f$-IRL (Ni et al., 2020). We believe that our methods are especially helpful on hard exploration problems (i.e. problems where many actions lead to low reward, while few, sparse actions lead to high reward, like in the toy tree MDP in Section 6). In these types of problems, it is difficult to rely on the learner to find high reward areas of the space through hard exploration, and informing the learner of expert states and actions through expert bootstrapping can significantly accelerate recovery of expert performance.

In summary, the main contributions of this paper are two recipes, ERB and EQB, which can be added onto state-of-the-art inverse reinforcement learning algorithms (with few lines of code) for accelerated learning. Empirically, we show that our techniques yield significant gains on the benchmark MuJoCo suite of tasks. In addition, we explain when and why our techniques are helpful through the study of a simple toy tree MDP.

## 2 RELATED WORK

### 2.1 LEVERAGING EXPERT DEMONSTRATIONS IN REINFORCEMENT LEARNING

Reinforcement learning algorithms (e.g. SAC (Haarnoja et al., 2018), DQN (Mnih et al., 2013), and PPO (Schulman et al., 2017)) aim to find an optimal policy by interacting with an MDP. Solving a reinforcement learning problem can require extensive exploration throughout a space to find potentially sparse reward. A standard practice is to bootstrap RL policies with a behavior cloning policy (Cheng et al., 2018; Sun et al., 2018). Deep Q Learning from Demonstrations(DQfD) (Hester et al., 2018) and Human Experience Replay (Hosu & Rebedea, 2016) propose to accelerate the exploration process by inserting expert transitions into the policy replay buffer in order to inform the learner of high reward states. However, these RL approaches assume access to stationary ground truth rewards, which is not the case in imitation learning where rewards are being learnt over time.

### 2.2 IMITATION LEARNING

Imitation learning algorithms attempt to find a policy that imitates a given set of expert demonstrations without access to ground truth rewards. Based on assumptions of available information, imitation learning algorithms can be broadly classified into three categories: offline (e.g. Behavior Cloning), interactive expert (e.g. DAgger (Ross et al., 2011), AggreVaTe (Ross & Bagnell, 2014)), or interactive simulator (e.g. MaxEntIRL (Ziebart et al., 2008)). While offline algorithms such as offline IQ-Learn (Garg et al., 2021) and AVRIL (Chan & van der Schaar, 2021) are sample efficient at leveraging expert data, they suffer from covariate shift due to the mismatch between expert and learner distributions. In this work, we use online inverse reinforcement learning algorithms to combat covariate shift and hence assume access to an interactive simulator.

Inverse reinforcement learning algorithms (e.g. MaxEntIRL (Ziebart et al., 2008), $f$-IRL (Ni et al., 2020)) attempt to recover a reward function that explains expert behavior. There are previous methods that aim at recovering a reward without solving an inner loop RL problem. For example, (Klein et al., 2012) assumed simple classes of reward functions; (Pirotta & Restelli, 2016; Ramponi et al., 2020) learned a reward function solely on expert states without considering the learner distribution. In contrast, we focus on general online inverse reinforcement learning methods. Adversarial imitation learning methods (e.g. AIRL (Fu et al., 2018), GAIL (Ho & Ermon, 2016)) attempt to find an optimal policy by using a discriminator instead of a reward function, and running policy optimization in the outer loop, as opposed to the inner loop in MaxEntIRL. Similar to our work, SQIL, or Soft-Q Imitation Learning (Reddy et al., 2019) inserts expert transitions into the learner replay buffer. However, SQIL only uses rewards of 0 for all learner transitions and rewards of 1 for all

expert transitions, which does not recover a reward function that can be used for other purposes, for example with different environments with different dynamics, and settings where the expert actions are not realizable (e.g. human hand vs. robot arm). IQ-Learn (Garg et al., 2021) proposes a new formulation which learns a single Q function to take the place of both the reward function and the SAC policy critic. IQ-Learn also inserts expert transitions into the replay buffer, for the purpose of estimating the expected policy value function $V^\pi$ on the initial state distribution $\rho_0$ and for policy optimization. However, the authors did not provide justification or empirical studies of this treatment. In addition, while the focus of the IQ-Learn paper is the new formulation, the focus of our paper is on general purpose recipes for using expert information to accelerate imitation learning.

## 3 BACKGROUND

The goal of reinforcement learning algorithms is to find a policy $\pi$ that maximizes reward in an MDP $(\mathcal{S}, \mathcal{A}, \mathcal{T}, r, \gamma, \rho_0)$, which is a tuple of states $\mathcal{S}$, actions $\mathcal{A}$, transition dynamics $\mathcal{T}$, reward function $r$, discount factor $\gamma$, and initial state distribution $\rho_0$. The policy is a mapping from states to a distribution over actions. The reward function is a mapping $r : \mathcal{S} \times \mathcal{A} \to \mathbb{R}$ from states and actions to a reward value. The goal of the policy is to maximize cumulative discounted reward on trajectories $\tau$ sampled from the trajectory distribution induced by policy $\pi$: $\max_\pi \mathbb{E}_{\tau \sim \pi} \left[ \sum_{t=0}^T \gamma^t r(s_t, a_t) \right]$.

In maximum entropy reinforcement learning, the goal is to find a policy $\pi$ that maximizes entropy-regularized reward: $\max_\pi \mathbb{E}_{\tau \sim \pi} \left[ \sum_{t=0}^T \gamma^t (r(s_t, a_t) + H(\pi(\cdot|s_t))) \right]$. The additional entropic regularization increases exploration by incentivizing the policy to spread out its probability mass over different actions instead of focusing its probability mass on a single action.

It can be shown that the optimal policy selects trajectories with probabilities proportional to the exponential of the ground truth cumulative discounted sum of rewards: $\pi^*(a|s) = \frac{1}{Z_s} \exp(Q^*(s, a))$, where $\pi^*$ denotes the optimal policy and $Q^*$ denotes the optimal action value function.

Soft-Actor Critic (SAC), a popular maximum entropy RL algorithm, alternates between optimizing a Q function, or critic, and optimizing a policy, or actor. SAC calculates target next state values for the Q updates on each transition by approximating the optimal action using an explicit policy $\pi$ and calculating the entropy-regularized Q value for this action in order to create the target (Equation 1). In Equation 1, $d$ is defined as a boolean 1 or 0 indicating whether the episode is over. For more details regarding this version of SAC, please refer to OpenAI Spinning Up (Achiam, 2018).

$$ y(r, s', d) = r + \gamma(1 - d)(\min_{i=1,2} Q_{\theta_{\text{targ}, i}}(s', \tilde{a}') - \alpha \log \pi_\theta(\tilde{a}'|s')), \tilde{a}' \sim \pi_\theta(\cdot|s') \tag{1} $$

The $\min_{i=1,2} Q_{\theta_{\text{targ}, i}}(s', \tilde{a}') - \alpha \log \pi_\theta(\tilde{a}'|s')$ term is essentially estimating $V^\pi(s')$, or the value on state $s'$ under the current policy. The objective of the Q function is

$$ \mathcal{L}(s, a, r, s', d) = (Q(s, a) - y(r, s', d))^2. \tag{2} $$

The policy's objective is to maximize the entropy regularized Q value

$$ \max_\pi \mathbb{E}_{a \sim \pi(\cdot|s)} \left[ Q(s, a) - \alpha \log \pi(a|s) \right]. \tag{3} $$

SAC is essentially alternating between policy evaluation, where the Q function is estimating the Q value under the current policy, and policy improvement, where the policy is trying to improve using the current Q function.

In imitation learning settings, the reward function $r$ is not given, and instead a set of expert demonstrations $\mathcal{D}$ is given. The goal in imitation learning is to find a policy that imitates the expert. In inverse reinforcement learning, this is done by recovering a reward function that explains expert behavior as shown in the expert demonstrations. In the maximum entropy formulation, the expert is assumed to be optimizing the entropy-regularized cumulative discounted sum of rewards. Inverse reinforcement learning algorithms such as $f$-IRL and MaxEntIRL alternate between taking one gradient step for the reward and solving the inner-loop RL problem to obtain the optimal policy with respect to the current reward function.

# 4 ERB: EXPERT REPLAY BOOTSTRAPPING

In algorithms such as MaxEntIRL, the policy typically ignores the expert data when solving the inner-loop RL problem, and instead must act solely based on the blackbox reward. We propose to accelerate the inner-loop RL problem by placing expert samples into the learner replay buffer and thereby *informing the learner about high reward expert transitions* instead of requiring the learner to explore a potentially vast space in order to reach the expert distribution. The detailed algorithm is described in Algorithm 1 with the additions to MaxEntIRL highlighted in red.

---

**Algorithm 1** ERB version of MaxEntIRL

---

**Input:** Expert demos $D_E$, epochs $N$, inner policy steps $T$
**Output:** Policy $\pi$, critic $Q$, reward function $r$
1: Initialize SAC policy $\pi$, critic $Q$, reward function $r$
2: Initialize learner replay buffer $D_L$
3: Initialize SAC policy learner replay buffer $D_P$
4: **for** $i = 0$ to $N$ **do**
5:   Collect policy rollouts $\tau_L$, place into $D_L$
6:   Sample expert and learner batches $B_E$ from $D_E$, $B_L$ from $D_L$
7:   Update reward with gradient
$$\mathbb{E}_{s,a \in B_E}\left[\nabla_{\theta_r} r(s,a)\right] - \mathbb{E}_{s,a \in B_L}\left[\nabla_{\theta_r} r(s,a)\right]$$
8:   **for** $j = 0$ to $T$ **do**
9:     Collect rollouts from $\pi$ and place into $D_P$
10:     Sample policy batch $B_P$ from $D_P$
11:     Sample expert batch $B_E$ from $D_E$
12:     $B_P = B_E \cup B_P$
13:     Use the objective defined in Equations 1 and 2 to update the critic $Q$
14:     Use the objective in Equation 3 to update the policy $\pi$
15:   **end for**
16: **end for**
17: **return** $\pi, r$

---

## 4.1 IS THIS JUST BEHAVIOR CLONING?

Behavior Cloning trains the learner to pick the expert action on expert states. However, there are several ways that Behavior Cloning may not function properly. For instance, the expert may not be realizable, in which case the learner may be led to off-expert-distribution states where it does not have any experience.

Instead, by providing the expert samples in the replay buffer, we allow the policy to choose *parts* of the expert transitions that are realizable and helpful in maximizing rewards even when the expert is not realizable as a whole. As a result, if the expert takes perfect actions while stochastic perturbations lead the learner to off-expert-distribution states, non-dynamics-aware BC will not be able to recover while dynamics-aware methods like MaxEntIRL will be able to. In a similar vein, there is strong evidence from the planning community (Phillips et al., 2012) that shows how expert demonstrations, even segments of it, can be used to construct useful heuristics to aid solving large MDPs.

We also evaluated Behavior Cloning and MaxEntIRL with ERB on a modified version of the Tree MDP presented in Section 6 where the learner has "shaky hands" and with 20% probability takes a random action at any given state instead of the learner's chosen action. In addition, one of the actions at any given state is to go back up to the current node's parent, as a means of recovering from the stochastic perturbations. Expert demonstrations are given showing the expert perfectly descending the tree on the left and without "shaky hands". In such a situation, BC suffers from covariate shift, as it is unable to recover after stochastic perturbations have led it to off-expert-distribution states, while dynamics-aware ERB is able to recover to the expert path and converge to higher returns, as shown in Figure 1(b).

## 5 EQB: EXPERT Q BOOTSTRAPPING

Standard SAC updates used in both reinforcement learning and imitation learning calculate target next state Q values for each transition by approximating the optimal action using an explicit policy $\pi$ and calculating the entropy-regularized Q value for this action in order to create the target (Equation 1).

However, the policy's approximation of the optimal action may be suboptimal, which can cause target value estimates to be significantly below what could be achieved by a stronger Q-optimizing policy. The lower targets cause the Q function to learn lower estimates of value and slows down learning. Ideally, the policy component of SAC would be able to strongly optimize the critic, and create a positive cycle where the policy provides better targets and accelerates the critic's progress, hence providing better signals to the policy. Our key observation is that when updating on expert states as in expert replay bootstrapping, the expert's near optimal next action is given and can be used to improve the target estimate. This extra information creates an opportunity to improve. *We propose to use a combination of the ground truth expert's next action and the policy's approximated action in order to create better targets for Q learning on expert states.*

Specifically, we modify the target equation on *expert states* to be

$$y_E(r, s', d) = r + \gamma(1-d)V_{\text{EQB}}(s', \tilde{a}', a'_E), \quad (s', a'_E) \in B_E, \tilde{a}' \sim \pi_\theta(\cdot|s') \tag{4}$$

$$Q_m(s, a) = \min_{i=1,2} Q_{\theta_{\text{targ},i}}(s, a) \tag{5}$$

$$V_{\text{EQB}}(s, \tilde{a}', a'_E) = \alpha \log(e^{\frac{Q_m(s, a'_E)}{\alpha}} + e^{\frac{Q_m(s, \tilde{a}')}{\alpha}}). \tag{6}$$

and the objective is thus

$$\mathcal{L}_{\text{EQB}}(s, a, r, s', d) = \begin{cases} (Q(s,a) - y_E(r, s', d))^2 & \text{if } s' \in B_E \\ (Q(s,a) - y(r, s', d))^2 & \text{otherwise} \end{cases}. \tag{7}$$

where $B_E$ denotes the set of expert transitions in our set of demonstrations. Note that $V_{\text{EQB}}$ is performing a softmax between two discrete actions, so it creates a mixture policy between the expert and learner selected actions.

Our final ERB+EQB learning algorithm consists of Algorithm 1 with Equations 4 and 7 defining the critic objective, replacing Equations 1 and 2 in line 13.

### 5.1 DERIVATION OF EQB Q TARGET

In maximum entropy inverse reinforcement learning with a SAC policy, the goal of the policy is to maximize the entropy regularized Q value (Equation 8).

$$\max_\pi \mathbb{E}_{a \sim \pi(\cdot|s)} [Q(s, a) - \alpha \log \pi(a|s)] \tag{8}$$

This policy is then used to select actions when rolling out in the environment and is used to estimate the target state value used to update the Q function (Equation 1). However, as the expert next action is given on expert states, the target value estimate can be improved with the additional information.

At state $s'$, suppose the policy suggests an action $\tilde{a}'$ and the expert suggests an action $a'_E$. We compose these two actions into a stronger meta policy, and more importantly, use the meta policy to derive value targets. The better value targets from the strong meta policy can accelerate the critic's progress and lead to better signals to the policy, therefore accelerating both sides and creating a virtuous cycle. We consider a class of meta policies $\Pi_M$ that select between the action chosen by the current policy and the given expert action. Each meta policy is governed by a probability $w$ such that the meta policy selects the current policy chosen action with probability $w$ and the expert next action with probability $1-w$. If we consider the policy and expert actions as discrete choices, then we can explicitly solve for the optimal value of $w$ for a meta policy $\pi_M$ that maximizes the *discrete* entropy-regularized Q value objective in Equation 8 over the two choices:

$$V_{\text{EQB}}(s) = \max_w [w(Q_m(s, \tilde{a}') - \alpha \log w) + (1-w)(Q_m(s, a'_E) - \alpha \log(1-w))] \tag{9}$$

$$= \alpha \log(e^{\frac{Q_m(s, a'_E)}{\alpha}} + e^{\frac{Q_m(s, \tilde{a}')}{\alpha}}). \tag{10}$$

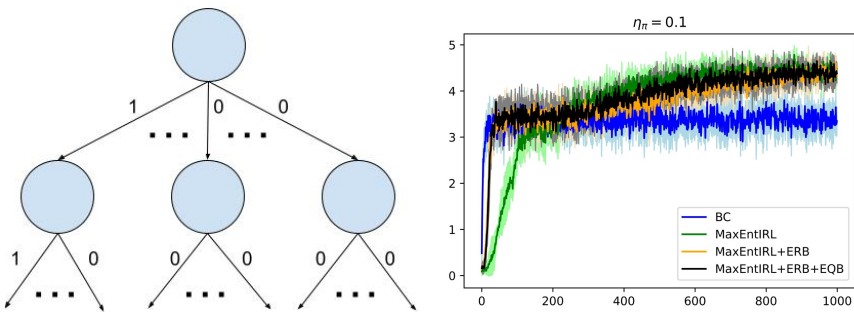

Figure 1: 1(a): An example toy tree MDP. Reward is only given for taking the leftmost actions. 1(b): Results for Behavior Cloning, MaxEntIRL, ERB, and EQB on the "shaky hands" MDP. Results are averaged over 5 seeds, and shaded areas represent standard deviation. The x-axis represents iterations, while the y-axis represents return.

More details on the derivation can be found in Appendix A.

## 6    INSIGHTS FROM A TOY PROBLEM

To gain more insight into why ERB and EQB are helpful, we evaluated ERB and EQB on a toy problem, consisting of an exponentially growing tree MDP with 7 levels. High reward is given for each action on the leftmost path of the tree. Expert demonstrations are given showing the expert taking only the leftmost path. We ran a simple SAC policy MaxEntIRL algorithm to evaluate the effectiveness of ERB and EQB. Though this problem is discrete, we used a separate policy from the Q function, as our methods are specifically designed for the continuous case. In a continuous setting, it is infeasible to explicitly maximize over the different action Q values at a state so we must have an additional separate policy that attempts to maximize the Q function. EQB is helpful when the separate policy is unable to effectively maximize the Q function. The policy, Q function, and reward function are all tabular. The policy $\pi$ is parameterized by values $v_\pi$, which determine a softmax distribution over actions. An example tree MDP is shown in Figure 1(a).

Our algorithm alternates between performing $n$ SAC updates, each of which consists of a Q update and a policy update, and performing 1 reward update. The original Q update is done using the following equation:

$$y = r + \gamma(1 - d)(Q(s', \tilde{a}') - \log(\pi(\tilde{a}'|s'))), \tilde{a}' \sim \pi(\cdot|s') \tag{11}$$
$$Q(s, a) := Q(s, a) + \eta_Q * (y - Q(s, a)). \tag{12}$$

For simplicity, the discount factor $\gamma$ and the entropy weight $\alpha$ are both set to 1. The EQB Q update on expert states is done using the modified target

$$y = r + \gamma(1 - d)(\log(e^{Q(s', \tilde{a}')} + e^{Q(s', a^*)})), \tilde{a}' \sim \pi(\cdot|s'). \tag{13}$$

The policy update is done by maximizing the expected entropy regularized Q value. The gradients are calculated using PyTorch Autograd.

$$\max_\pi \mathbb{E}_{a \sim \pi(\cdot|s)} \left[ Q(s, a) - \log \pi(a|s) \right]. \tag{14}$$

The update is then done:

$$v_\pi := v_\pi + \eta_\pi * \nabla_{v_\pi} \mathbb{E}_{a \sim \pi(\cdot|s)} \left[ Q(s, a) - \log \pi(a|s) \right]. \tag{15}$$

The reward update simply increases reward on expert states and decreases reward on learner states:

$$r(s, a) := r(s, a) + \eta_r * (\mathbb{1}((s, a) \text{ is expert})) - \eta_r * (\mathbb{1}((s, a) \text{ is learner})). \tag{16}$$

In the experiments in Figure 2, learning rates $\eta_Q$ and $\eta_r$ are both set to 0.01, while $\eta_\pi$ is set to varying values in $\{0.01, 0.001, 0.0001\}$. Here $\eta_\pi$ serves as an optimization difficulty knob that controls how well the policy can optimize the Q function.

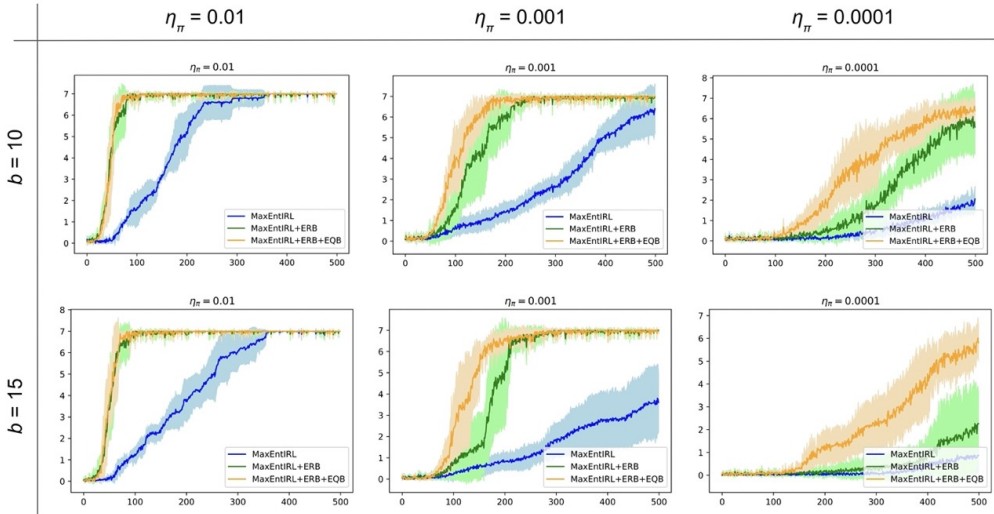

Figure 2: Top: Results for toy tree MDP task with branching factor 10 and $\eta_\pi$ set to different values. Results averaged over 5 seeds, shaded areas represent standard deviation. Bottom: Results for toy tree MDP task with branching factor 15. X-axis represents iterations, y-axis represents return.

We also tested using a branching factor $b = 10$ versus 15 in our experiments. A larger branching factor creates a harder exploration problem, as there are sparser rewards. This creates more room for improvement from ERB and EQB, as they both cut down on exploration by giving the policy expert information and allow the policy to quickly find expert regions. As shown in the bottom half of Figure 2, ERB and EQB yield bigger improvements when the branching factor is larger.

**Why is ERB helpful?** ERB is helpful in accelerating learning as instead of relying on the policy to explore the exponential tree and find high reward states, it directly informs the learner of high reward expert states and accelerates learning.

**Why is EQB helpful?** EQB allows the Q function to progress in its learning even when the policy may provide suboptimal target estimates. The policy may be unable to effectively optimize the Q function even though the Q function may be accurately estimating high value in expert transitions. In this case, using the expert action to estimate target value as in EQB allows for an accurate target value estimate that is not incorrectly low because of the suboptimality of the policy. Using this target value estimate, the Q function is able to progress in its learning and continue to provide more accurate signals to the policy, instead of being held back by the suboptimal policy, hence accelerating learning. This effect is illustrated in the horizontal variation in Figure 2, where we vary the learning rate of the policy to allow for different degrees of maximization of the Q function to measure the effectiveness of EQB. When the policy is able to effectively maximize the Q function, the effect of EQB diminishes. On the other hand, when the policy is unable to effectively maximize the Q function, EQB significantly accelerates learning. Though using only the expert's next action to calculate target values is a valid approach, we believe that using a combination between the expert's next action and the policy's chosen action is necessary. As the Q function may be erroneous during training, and the policy is explicitly optimizing the Q function, using the policy's chosen action can help catch erroneously high Q values and correct them as needed. As there are also cases in which the policy's chosen action is suboptimal, we believe that using a mixture of the policy and expert Q values defends against either extreme.

The modified target value estimate is no longer $V^\pi$ as in SAC, it is estimating target value for the meta policy $\pi_M$. We believe that this is beneficial as it more accurately estimates the high value in expert states and moves the learner towards expert states, which is the ultimate goal.

## 7    EXPERIMENTS

We evaluated our methods on the benchmark MuJoCo (Todorov et al., 2012) suite of tasks in OpenAI Gym (Brockman et al., 2016), building on top of MaxEntIRL and $f$-IRL. We built on top of an open-

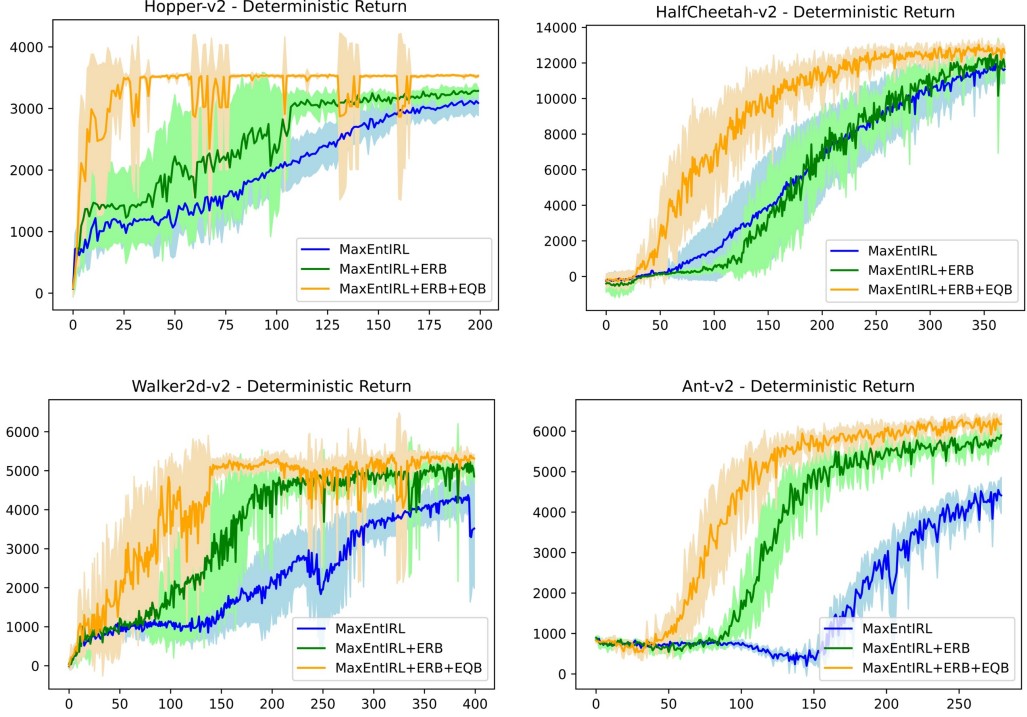

Figure 3: Deterministic returns on 4 MuJoCo tasks with a MaxEntIRL baseline. X-axis is iterations, where each iteration is 5000 policy learning environment steps.

source implementation (that calculates state-only rewards to disentangle reward from dynamics): https://github.com/twni2016/f-IRL.git and added expert replay bootstrapping and expert Q bootstrapping as described in Section 4 and Section 5. In all experiments, we report results averaged over 5 seeds, and shaded areas represent standard deviation. In all plots, the x-axis is iterations, and each iteration is 5000 policy learning environment steps, and the y-axis is return. All hyperparameters remain the same as in the original $f$-IRL configs except for the entropy weight $\alpha$ on expert state EQB Q updates due to the difference in the scale of entropy in EQB. More details are given in Appendix B. All experiments with expert replay bootstrapping are run with batches of half expert samples and half learner samples. We study the effect of different ratios of expert samples in Appendix C. The total number of samples used in each batch for updates remains the same.

**MaxEntIRL Baseline:** We report results on the four benchmark MuJoCo environments reported in the $f$-IRL paper with a MaxEntIRL baseline. Performance graphs showing deterministic return are given in Figure 3, and performance graphs showing stochastic return are given in Figure 5. Expert performance is reported in Table 2. Each unit on the x-axis represents 5000 policy learning environment steps, though there is an additional 10000 learner rollout steps required by the reward function to update the reward replay buffer of learner states. We further report the amount of iterations to recover 50%, 70%, and 90% of deterministic expert performance in Tables 3, 1, and 4 respectively. It is seen that our methods show significant gains over a MaxEntIRL baseline on the benchmark MuJoCo suite of tasks, speeding up recovery to 70% of deterministic expert performance by 2.13x on HalfCheetah-v2, 2.6x on Ant-v2, 18x on Hopper-v2, and 3.36x on Walker2d-v2.

**$f$-IRL (Ni et al., 2020) Baseline:** We also evaluated our methods on top of 3 variants of an $f$-IRL baseline, minimizing Forward KL, Reverse KL, and Jensen-Shannon Divergence. Detailed results are given in Appendix D. We found that the 3 variants achieved similar performance with each other and with the MaxEntIRL baseline. We found that ERB and EQB gave consistent, significant improvement on the Reverse KL and Jensen-Shannon Divergence variants for all four environments, and on the Forward KL variant of $f$-IRL, ERB and EQB gave significant improvement in terms of real return on 3 of the 4 environments. As an example, Deterministic Return for ERB and EQB on top of the Jensen-Shannon variant of $f$-IRL is given in Figure 4. However, on Walker2d with a

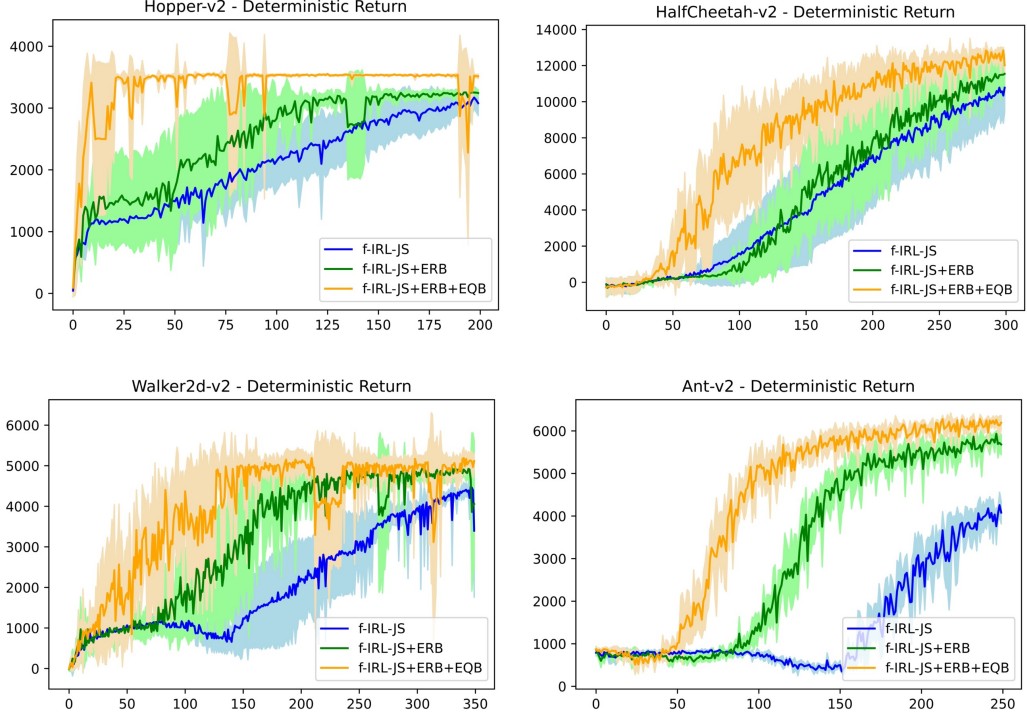

Figure 4: Deterministic returns on 4 MuJoCo tasks with a Jensen-Shannon Divergence $f$-IRL baseline. Each unit on the x-axis represents one iteration, or 5000 policy update environment steps.

Table 1: Number of iterations to recover 70% of deterministic expert performance. Lower is better.

| Task | Original | ERB | ERB+EQB |
|------|----------|-----|---------|
| Hopper-v2 | 126 | 83 | 7 |
| HalfCheetah-v2 | 245 | 234 | 115 |
| Walker2d-v2 | 296 | 160 | 88 |
| Ant-v2 | 242 | 129 | 93 |

Forward KL $f$-IRL baseline, EQB yielded improvement on the surrogate Forward KL objective that $f$-IRL-FKL is optimizing, but did not do well in terms of real return. We suspect this is due to a disconnect between Forward KL and return on Walker2d. Detailed discussion is given in Appendix E.

## 8 CONCLUSION

In summary, we have presented two methods for accelerating inverse reinforcement learning algorithms (e.g. $f$-IRL, MaxEntIRL) that are generally applicable to any such IRL algorithms with an off-policy RL algorithm serving as the learner. Our first method, expert replay bootstrapping, consists of placing expert transitions into the learner replay buffer and performing learner updates on these transitions. We believe that this informs the learner of high value expert states instead of relying on hard exploration to reach expert states. Our second method, expert Q bootstrapping, consists of using the expert next action to create a better target value estimate on expert states. This results in more accurate estimates of value on expert states instead of potentially suboptimal policy action approximations that lead the learner away from expert states. Empirically, our methods significantly accelerate recovery to expert performance on the benchmark MuJoCo suite of tasks. Promising future directions include building stronger theoretical foundations for expert bootstrapping and exploring higher dimensional MDPs that can benefit from acceleration.

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

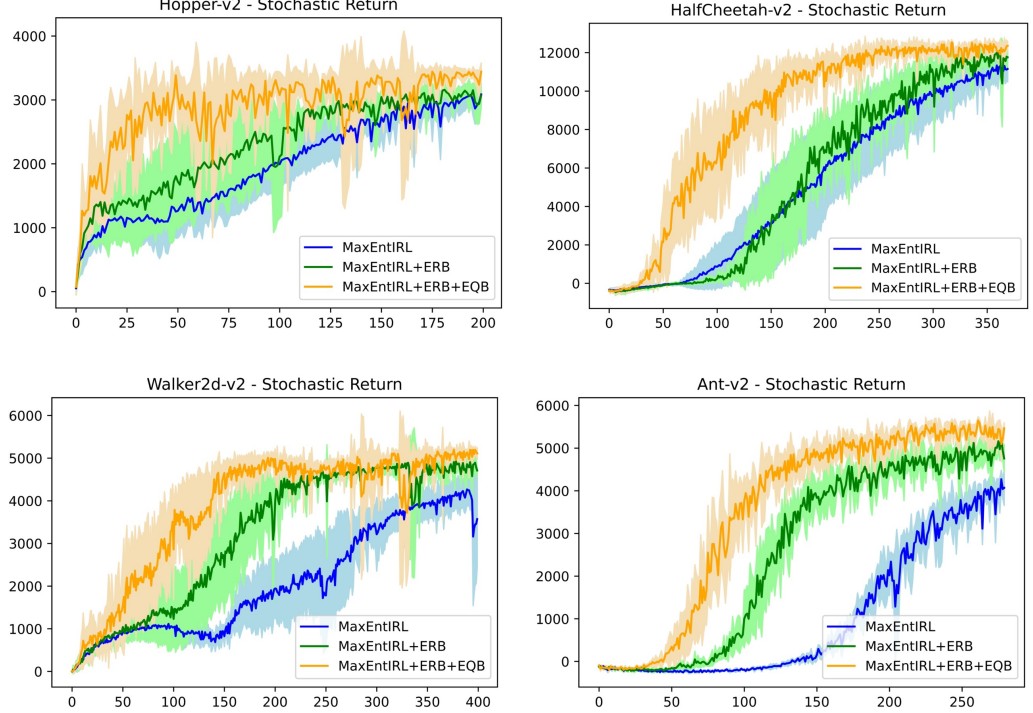

Figure 5: Stochastic returns on 4 MuJoCo tasks with a MaxEntIRL baseline. X-axis is iterations, where each iteration is 5000 policy learning environment steps.

Table 2: Expert Performance for MaxEntIRL. Higher is better.

| Task | Average Demonstration Performance |
|---|---|
| Hopper-v2 | $3497.89 \pm 10.70$ |
| HalfCheetah-v2 | $12559.94 \pm 169.32$ |
| Walker2d-v2 | $5283.72 \pm 62.70$ |
| Ant-v2 | $5928.87 \pm 136.44$ |

## A    DETAILS OF SECTION 5.1

Substituting into the objective in Equation 8 using the meta policy $\pi_M$ gives us

$$V^{\pi_M}(s) = V_{\text{EQB}}(s) = \max_{\pi_M} \mathbb{E}_{a \sim \pi_M(\cdot|s)} \left[ Q_m(s,a) - \alpha \log \pi_M(a|s) \right], \tag{17}$$

where the probability of selecting an action is simply the discrete probability of choosing the action multiplied by the continuous density of that action under the original policy (we approximate the expert's density with the current policy).

$$\pi_M(a|s) = w * \pi(a|s) \tag{18}$$

As the only parameter of the policy is the probability $w$ of selecting the current policy's chosen action, we can expand the expectation and obtain

$$V_{\text{EQB}}(s) = \max_{w} [w(Q_m(s, \tilde{a}') - \alpha \log \pi(\tilde{a}'|s) - \alpha \log w) +$$
$$(1-w)(Q_m(s, a'_E) - \alpha \log \pi(a'_E|s) - \alpha \log(1-w))]. \tag{19}$$

where $\tilde{a}'$ and $a'_E$ are the current policy's chosen action and the ground truth expert next action respectively, and both are given. We will denote $\tilde{a}'$ and $a'_E$ by $x_1$ and $x_2$ respectively in the following to simplify notation.

Table 3: Number of iterations to recover 50% of deterministic expert performance. Lower is better.

| Task | Original | ERB | ERB+EQB |
|------|----------|-----|---------|
| Hopper-v2 | 84 | 41 | 4 |
| HalfCheetah-v2 | 190 | 183 | 80 |
| Walker2d-v2 | 221 | 134 | 56 |
| Ant-v2 | 201 | 114 | 75 |

Table 4: Number of iterations to recover 90% of deterministic expert performance. Lower is better.

| Task | Original | ERB | ERB+EQB |
|------|----------|-----|---------|
| Hopper-v2 | >200 | 140 | 19 |
| HalfCheetah-v2 | 331 | 299 | 180 |
| Walker2d-v2 | >438 | 207 | 98 |
| Ant-v2 | >284 | 165 | 121 |

Setting the derivative equal to 0, we obtain

$$Q_m(s, x_1) - \alpha \log \pi(x_1|s) - \alpha(\log w + 1)$$
$$- Q_m(s, x_2) + \alpha \log \pi(x_2|s) + \alpha(\log(1-w) + 1) = 0, \tag{20}$$

which gives us

$$w = \frac{1}{1 + e^{\frac{(Q_m(s,x_2) - \alpha \log \pi(x_2|s)) - (Q_m(s,x_1) - \alpha \log \pi(x_1|s))}{\alpha}}}. \tag{21}$$

Substituting this into the original formula gives us

$$V_{\text{EQB}}(s) = \alpha \log(e^{\frac{Q_m(s,\tilde{a}') - \alpha \log \pi(\tilde{a}'|s)}{\alpha}} + e^{\frac{Q_m(s,a'_E) - \alpha \log \pi(a'_E|s)}{\alpha}}). \tag{22}$$

At convergence, taking into account the continuous density, EQB adds a constant $\log 2$ term to the $V^\pi(s')$ term in Equation 1. In practice, we found that removing the additional entropy term in the exponent was helpful to performance, resulting in Equation 10.

## B EXPERT STATE Q UPDATE ENTROPY WEIGHT IN EQB

As the modified EQB target value equation does not factor in entropy from the continuous action selection, and treats the expert and policy selected actions as given, the scale of the discrete entropy on expert states is different when performing EQB Q updates. Thus, additional tuning may be required to find the correct $\alpha$ entropy weight when applying EQB in expert state Q updates. We found that using an entropy weight of $\alpha = 1$ in value estimation on expert state Q updates in EQB (Equation 6) worked better than the original $\alpha = 0.2$. However, using $\alpha = 1$ in expert state Q updates on ERB did not reach EQB level performance and in some cases even significantly hurt performance, as shown in Figures 6 and 7, indicating that the resulting gain from EQB is not from hyperparameter tuning. All experiments use an entropy weight of $\alpha = 1$ for EQB on expert state Q updates unless specified otherwise.

## C DIFFERENT ERB RATIOS

We experimented with various ratios between expert and learner samples in order to study the effect of different amounts of expert versus learner information on imitation performance. Our results indicate that using 0 expert samples and using 0 learner samples perform worse than the default of using half expert, half learner samples. The latter is likely due to the absence of a normalizing force that lowers the value on learner states. As the policy is only given access to high reward expert

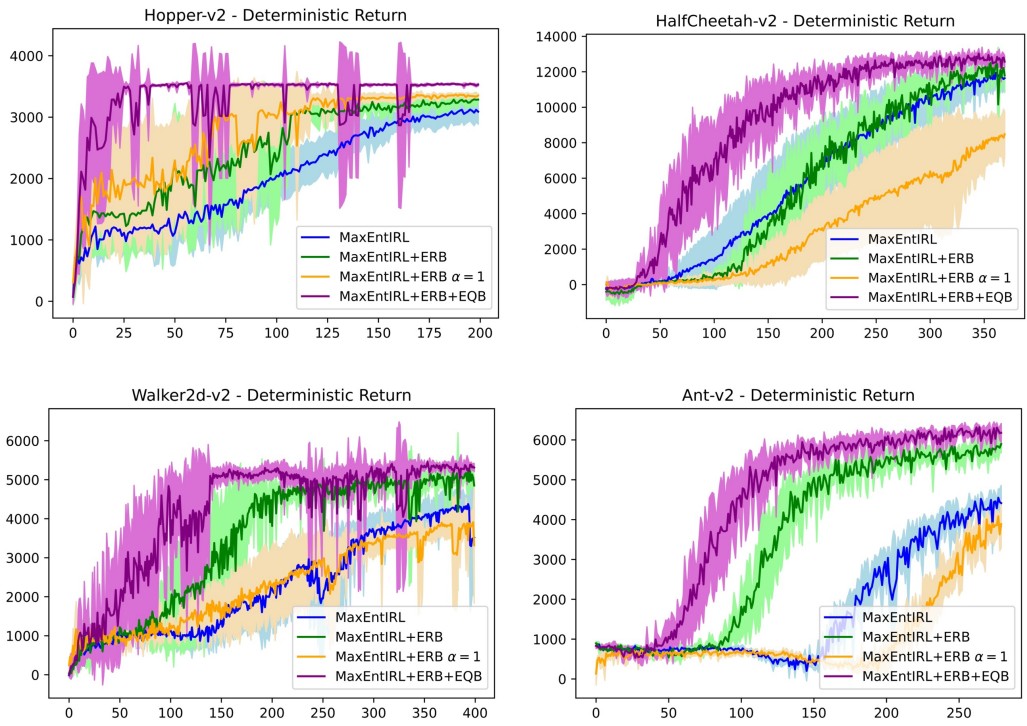

Figure 6: Deterministic returns for ERB with $\alpha = 1$ compared to other methods.

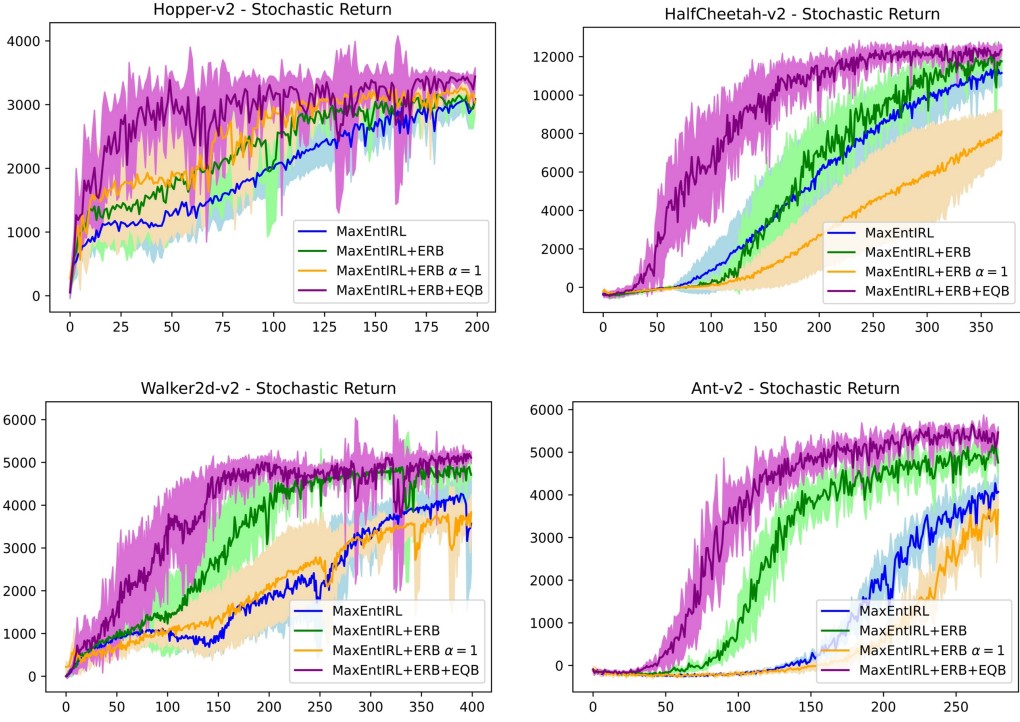

Figure 7: Stochastic returns for ERB with $\alpha = 1$ compared to other methods.

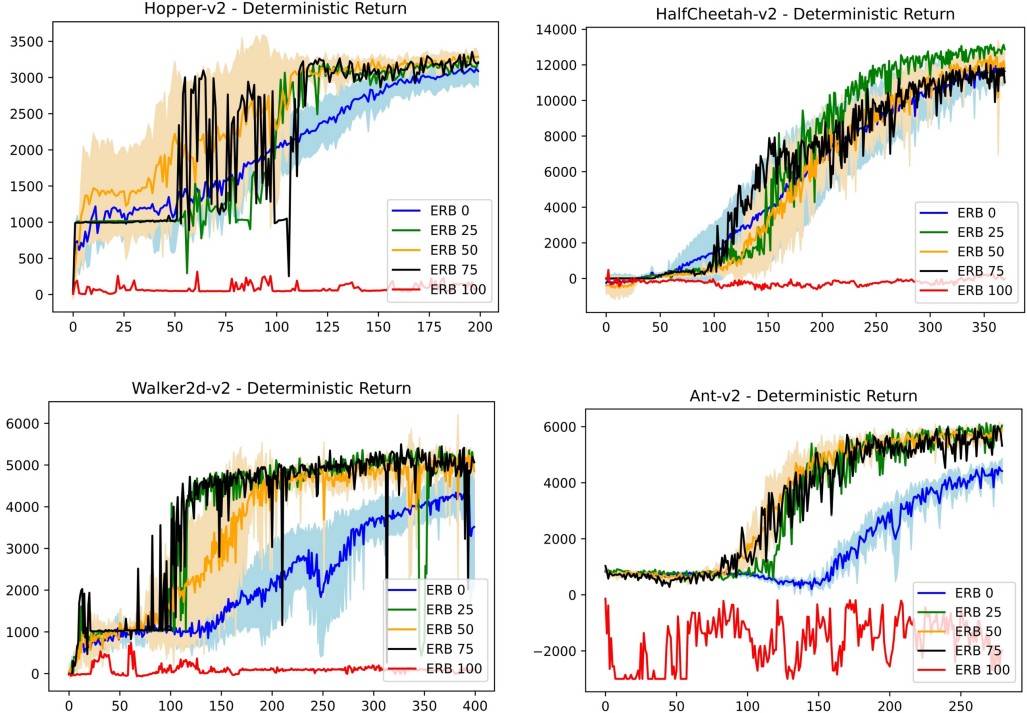

Figure 8: Deterministic returns for MaxEntIRL+ERB with different ratios of expert samples. In the figure, ERB X denotes that X percent of update transitions in the expert replay buffer were expert transitions.

transitions, there is no force to decrease predicted value in non-expert areas, and all values will naturally float up. Learner samples serve the purpose of providing a normalization force to make sure that non-expert areas do not have incorrectly high value. In addition, no particular non-extreme ratio of expert samples out of $\{25, 50, 75\}$ is consistently better or worse than the others. Hence, we recommend using a default ratio of 50% expert samples in order to strike a balance between learner and expert information. Deterministic returns for this study are shown in Figure 8, and stochastic returns are shown in Figure 9.

## D   $f$-IRL RESULTS

We evaluated ERB and EQB on three variants of an f-IRL baseline, minimizing Forward Kullback-Leibler, Reverse Kullback-Leibler, and Jensen-Shannon Divergence. For the Forward KL variant, Deterministic Returns are given in Figure 10, and Stochastic Returns are given in Figure 11. For the Reverse KL variant, Deterministic Returns are given in Figure 12, and Stochastic Returns are given in Figure 13. For the Jensen-Shannon variant, Deterministic Returns are given in Figure 14, and Stochastic Returns are given in Figure 15.

## E   WALKER2D $f$-IRL-FKL RESULTS

When evaluating our methods ERB and EQB on an $f$-IRL-FKL baseline, we found that both ERB and EQB provided consistent improvement on both Forward KL and real return over the $f$-IRL baseline on 3 of the 4 environments. However, on Walker2d, EQB provides improvement on the Forward KL surrogate objective but not on the return. We suspect this is due to the mismatch between the surrogate KL objective that $f$-IRL is optimizing and the return on Walker2d, while on other environments the two are highly correlated. Deterministic returns are given in Figure 10, while Forward KL is given in Figure 16.

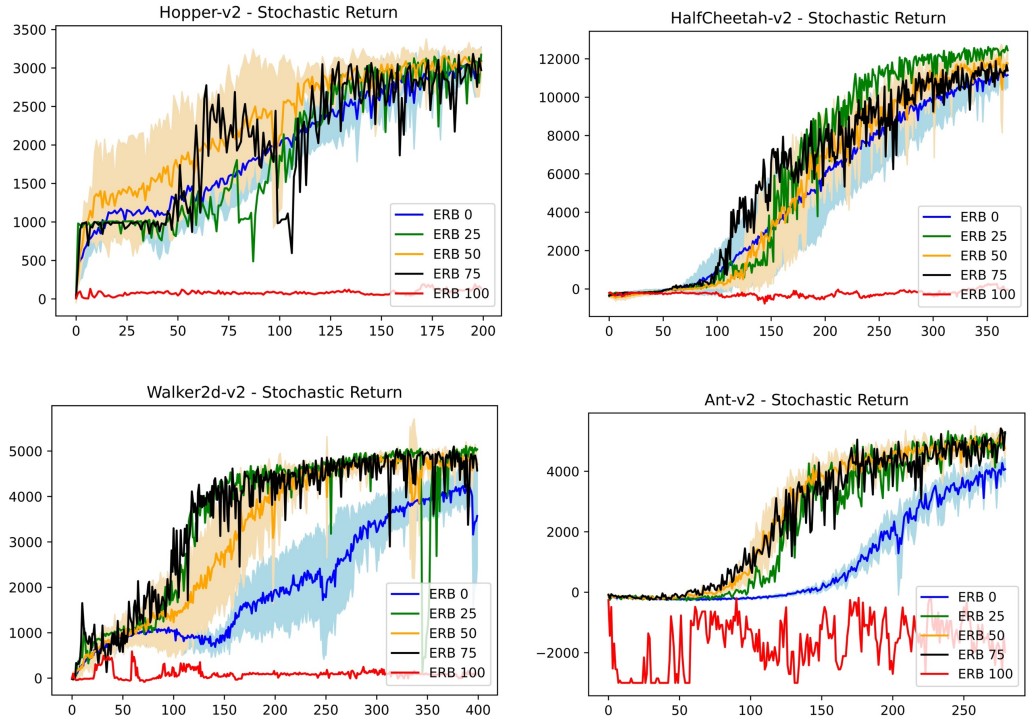

Figure 9: Stochastic returns for MaxEntIRL+ERB with different ratios of expert samples. In the figure, ERB X denotes that X percent of update transitions in the expert replay buffer were expert transitions.

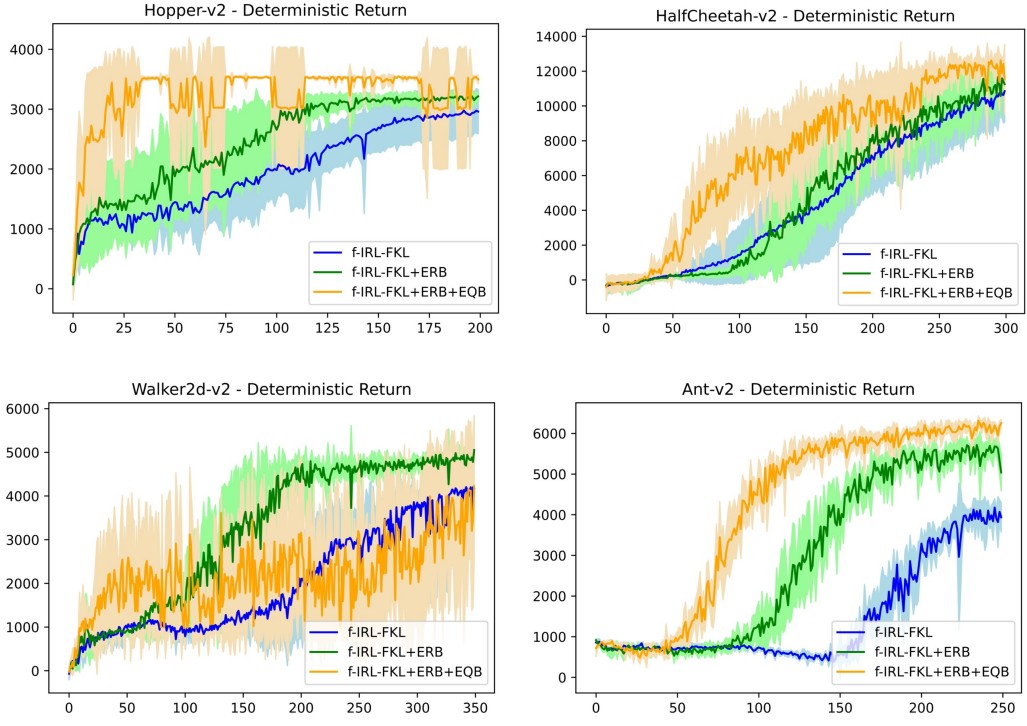

Figure 10: Deterministic returns on 4 MuJoCo tasks with a Forward KL Divergence $f$-IRL baseline. Each unit on the x-axis represents one iteration, or 5000 policy update environment steps.

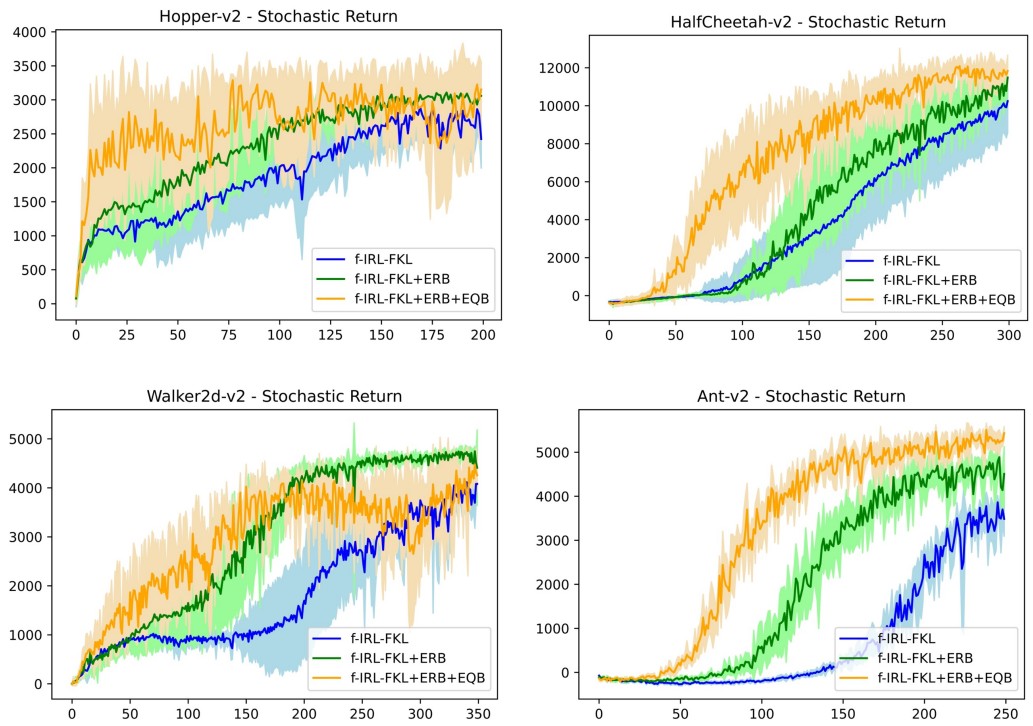

Figure 11: Stochastic returns on 4 MuJoCo tasks with a Forward KL Divergence $f$-IRL baseline. Each unit on the x-axis represents one iteration, or 5000 policy update environment steps.

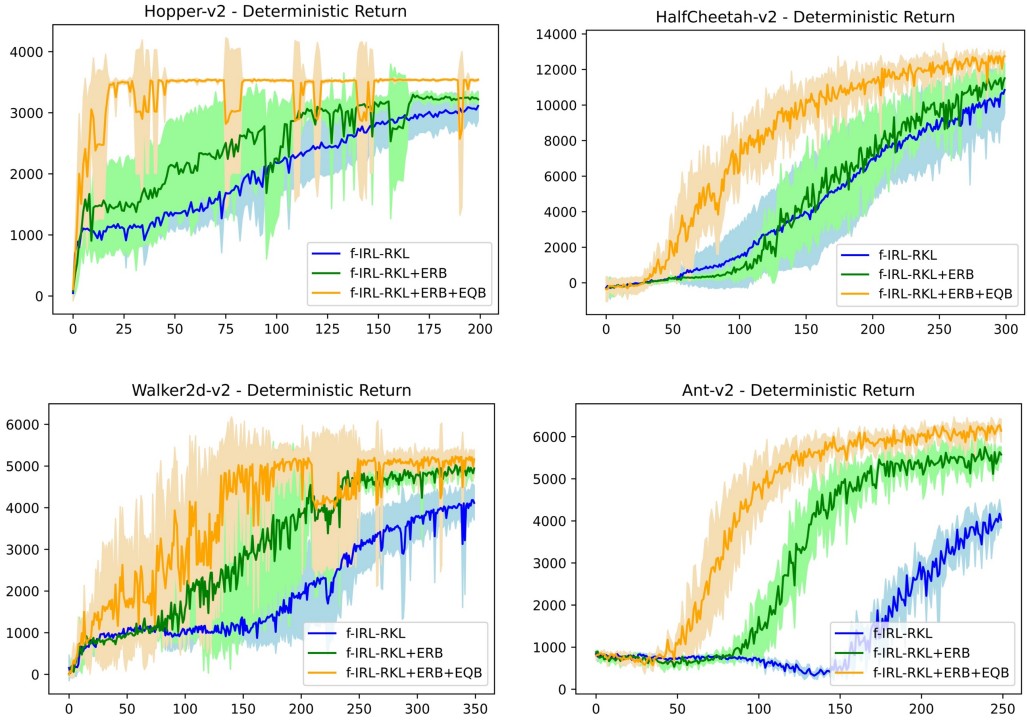

Figure 12: Deterministic returns on 4 MuJoCo tasks with a Reverse Kullback-Leibler $f$-IRL baseline. Each unit on the x-axis represents one iteration, or 5000 policy update environment steps.

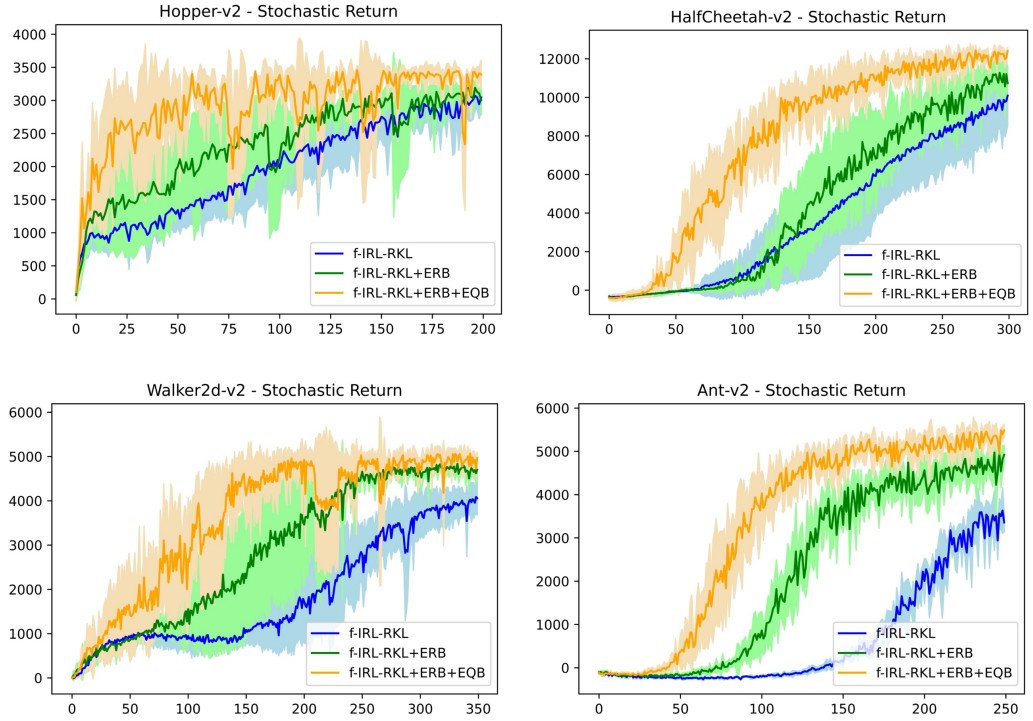

Figure 13: Stochastic returns on 4 MuJoCo tasks with a Reverse Kullback-Leibler Divergence $f$-IRL baseline. Each unit on the x-axis represents one iteration, or 5000 policy update environment steps.

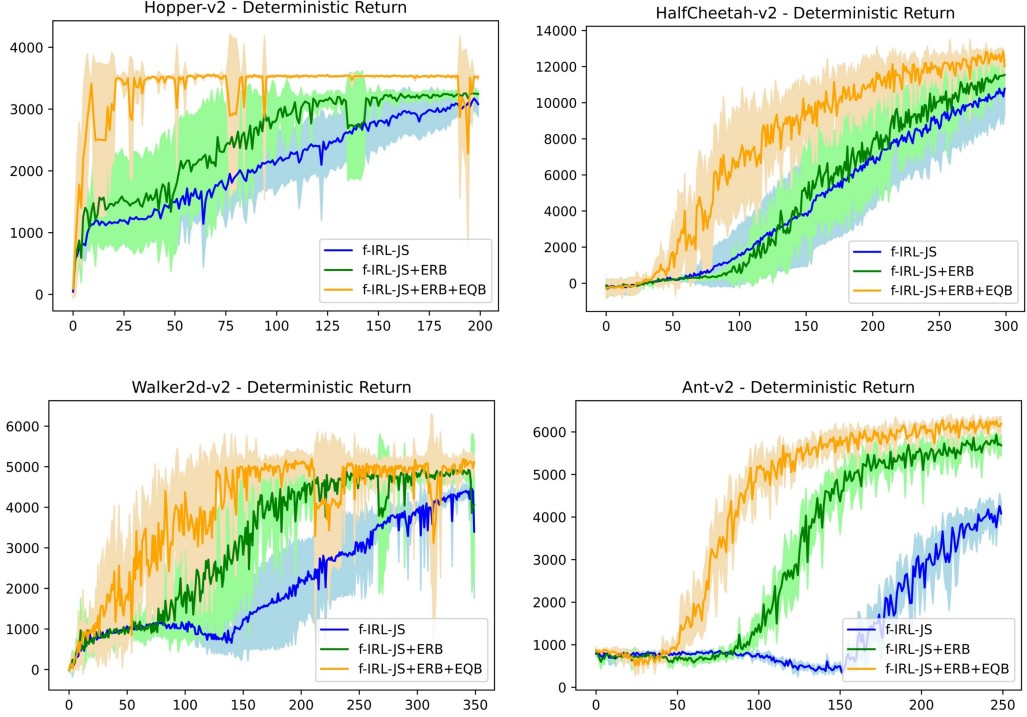

Figure 14: Deterministic returns on 4 MuJoCo tasks with a Jensen-Shannon Divergence $f$-IRL baseline. Each unit on the x-axis represents one iteration, or 5000 policy update environment steps.

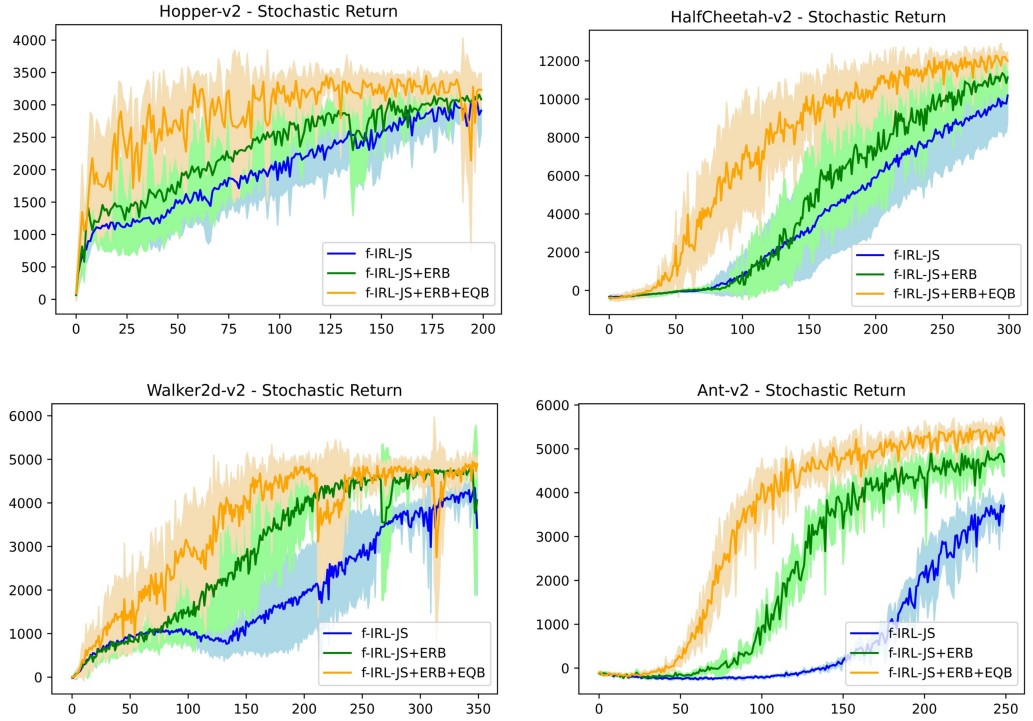

Figure 15: Stochastic returns on 4 MuJoCo tasks with a Jensen-Shannon Divergence $f$-IRL baseline. Each unit on the x-axis represents one iteration, or 5000 policy update environment steps.

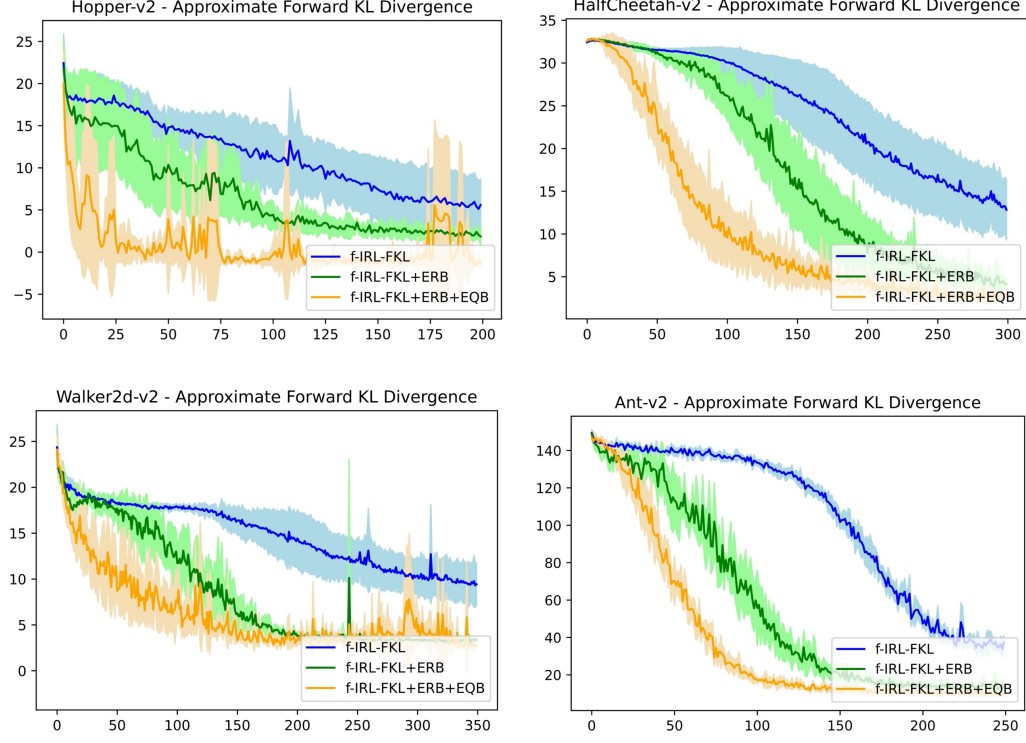

Figure 16: Approximate Forward KL Divergence over time on a Forward KL $f$-IRL baseline. Each unit on the x-axis represents one iteration, or 5000 policy update environment steps.

