# OpenReview forum: "Accelerating Inverse Reinforcement Learning with Expert Bootstrapping"
_ICLR.cc/2023/Conference — Submitted to ICLR 2023_

### Official Review · Reviewer_K4n8 · 2022-10-16

**Confidence:** 4
**Correctness:** 3
**Technical Novelty And Significance:** 2
**Empirical Novelty And Significance:** 2
**Recommendation:** 3

**Clarity, Quality, Novelty And Reproducibility:**

As I mentioned above, the paper is well-structured and easy to follow. This paper has found interesting research questions, but the proposed methods resemble the ideas of previous work, more importantly, the research question is not properly answered by the current version of this paper. Some issues have not been resolved, and there are no theoretical results to quantify the decrease in sample complexity by adding ERB and EQB. Since this is an important research question, the empirical results are insufficient. I strongly recommend continuing this research by considering the points I mentioned above. In terms of reproducibility, I believe the proposed algorithm can be easily implemented.

**Strength And Weaknesses:**

**Strength**

1. This paper focuses on an intriguing research question. To the best of my knowledge, while many previous IRL works did apply the expert demonstrate in policy update, but they have not carefully stuidied this method but rather treat it as an engineering trick that could potentially increase the training speed. I agree with the authors that this method requires a more carefully analyze by demonstrating its impact empirically and theoretically. While this paper does touch some of these aspects, I believe a more carefully study is required (please see my comment below).
2. The paper is well-structured and easy to follow with sufficient details for supporting understanding.

**Weaknesses**
I suggest the author to improve their paper by considering the following ideas:

1. **The Size of Policy Update**. In order to prevent destructively large policy updates, many RL algorithms, including the most popular PPO and TRPO algorithms, often include a constraint on the size of the policy update. They carefully tuned the objective to exclude state-action pairs from the policy that deviates from the current policy, but in this work, both ERB and EQB directly incorporate expert demonstration into the policy update and **the expert policy is significantly different from the imitation policy, especially in the beginning of training, which contradicts to the motivation of PPO or TRPO and is likely to drive a sub-optimal policy**. It is why MaxEnt IRL outperforms the proposed model in Figure 1(b) and Figure 2. I strongly recommend referring to the idea of constraining updates.

2. **Offline IRL baselines** Depending on whether IRL requires interaction with the environment, we can divide IRL into an online version and an offline version. For the Offline IRL algorithms (including IQ-Learn which has both the online and offline versions and AVRIL[1],), their policy update relies only on the expert demonstration data, so they require no more help from ERB and EQB mentioned in this paper. **I believe the author should include these offline IRL methods in their related works, and they could serve as strong baselines in this paper**.

[1] Alex J. Chan and Mihaela van der Schaar. Scalable Bayesian Inverse Reinforcement Learning. ICLR 2021.

3. **Testing Environment** The MoJoCo benchmark has 8 environments that are commonly applied for evaluation, but this paper studies only 4 of them. I am wondering how well ERB and EQB perform in the rest of the environments.

4. **Additional Concerns**

- Please include labels into the x-y axis of all plots. The plots without labels are difficult to read.

- "*which does not recover a reward function that can be used for other purposes.*"
I am confused by the term "other purposes". Please clarify.

- "*Formula (7)*"
I am wondering how to measure if $s^{\prime}$ is included in $B_{E}$. Will EQB examines all the samples and all the features to find the exact match? If yes, this step will be extremely time-consuming. I suggest training a density model for measuring the density of the input state in the expert distribution. If this density is large enough, we can use the EQB loss.

- "*Each meta policy is governed by a probability w such that the meta policy selects the current policy chosen action with probability w and the expert next action with probability 1 − w.*"
I am confused by the definition of $\omega$, especially when the authors substitute $log\pi$ with $log\omega$ in Appendix A. I think $\omega$ is an important weight between the current and the expert policy, and intuitively, the agent should simply pick the expert action to maximize the Q values, which contradicts Formula (10).

- MaxEnt IRL outperforms ERB and EQB in Figure 1 (b) and Figure 2, although it consumes more episodes or steps. I have explained this issue from the perspective of constraining updates.

**Summary Of The Paper:**

This paper studies an intriguing research question: utilizing the expert demonstration during the policy update step in an Inverse Reinforcement Learning (IRL) algorithm. This technique has been applied by many previous works, but none of them has carefully analyzed the impact of including expert demonstration. In order to utilize the demonstration data, the authors mainly focus on two methods: 1) adding expert transitions into the replay buffer of the inner RL algorithm. 2) using expert actions in Q value bootstrapping in order to improve the target Q value estimates and more accurately describe high-value expert states. The paper introduces the motivation and intuitively analyses the impact of each of these methods. In the experiment, the paper use MaxEnt-IRL and f-IRL as baselines for demonstrating the increase in training speed after including expert demonstration.

**Summary Of The Review:**

This paper describes a very interesting research question by studying the approach of including expert data in the policy update of IRL algorithms. The proposed ERB and EQB algorithms are intuitive and easy to follow, but it requires a more careful understanding of the impact of using expert data, especially when they are significantly different from the current policy. The over-aggressive update can cause many issues (Check the motivation of the update constraining methods like TRPO and PPO). The comparison is incomplete without comparing the offline IRL methods. Overall, I am inclined to reject the paper based on its current version, but I agree the research question is important and I believe the authors can resolve these issues by continuing this research.

---

> ### Author Response · Authors · 2022-11-11
> **Response to Reviewer K4n8**
>
> We thank the reviewer for the valuable time and feedback. We address the concerns below:
>  - **“The Size of Policy Update”**: On policy constraints: We note that SAC does not require constraints on the policy because only policy gradient methods and on-policy RL algorithms (e.g. PPO and TRPO) require the constraints as the update must be done using policies that are similar. Constraints are critical in on-policy policy gradient algorithms, as the data needs to be from the current policy, but are not required in off-policy algorithms like SAC, DDPG, and TD3, where the update transition data does not even need to be from the policy. From the SAC paper [1]: “both value estimators and the policy can be trained entirely on off-policy data.” In addition, the entropic regularization prevents overly-aggressive updates.
>  - **“It is why MaxEnt IRL outperforms the proposed model in Figure 1(b) and Figure 2.”**: We note that ERB and EQB yield significant gains over MaxEntIRL in both Figures 1b and 2: In Figure 1b, ERB and EQB allows much faster recovery of large chunks of expert performance during the beginning of training, while Figure 2 demonstrates that ERB and EQB (green and orange) yield clear improvements in both speed and performance over MaxEntIRL (blue) on all 6 hyperparameter settings.
>  - **“Offline IRL baselines”**: We thank the reviewer for the suggestion and will include the mentioned Offline IRL methods in our related work. These methods allow for effective use of expert data, though they suffer from covariate shift, as the learner rollout distribution does not match the expert distribution on which updates are done. In this paper, we assume the online setting. Our methods provide improvements over online IRL methods, which mitigate the issue of covariate shift while still effectively leveraging expert data.
>    - Empirically, our preliminary results indicated that IQ-Learn suffers from instability issues, similar to the observations of [2]. (See our “IQ-Learn Results” on our supplementary figures website [here](https://sites.google.com/view/airleb-supp-figures)) We will add additional discussion regarding offline IRL methods, including IQ-Learn in our Related Work section.
>    - AVRIL and other offline IRL methods suffer from covariate shift, as the learner rollout distribution does not match the expert distribution on which updates are performed. In addition, as demonstrated in the AVRIL paper (Fig. 3 in the paper), AVRIL is unable to consistently improve over BC on the three MuJoCo environments where comparisons are done, while f-IRL[3] consistently demonstrates the superior performance of MaxEntIRL and f-IRL over BC and other imitation learning baselines.
>  - **“Please include labels into the x-y axis of all plots.”**: We will add explanations to the x-y axes on our graphs.
>  - **“other purposes”**: Learned reward functions through IRL can be applied to different environments with different dynamics, and settings where the expert actions are not realizable (e.g. human hand vs. robot arm) as they capture the goals of the agent.
>  - **“Formula (7)”**: We obtain samples of $s’$ and $a’$ from the expert transition $(s, a, r, s’, a’)$. This does not require checking all samples and features as the expert’s next action is passed directly with expert transitions. However, if we wanted a low variance estimate, training a density model is also a good suggestion.
>  - **“Each meta policy is governed by a probability w such that the meta policy selects the current policy chosen action with probability w and the expert next action with probability 1-w.”**: $\omega$ is the probability that the meta policy selects the expert action (hence $\pi(a|s)$). The meta policy does not directly maximize Q value, but instead maximizes entropy-regularized Q value, which is part of the maximum entropy formulation of MaxEntIRL. In a non-maximum-entropy formulation, the meta policy may instead directly select the maximum Q value for learner and expert actions.
>
> [1] Tuomas Haarnoja, Aurick Zhou, Pieter Abbeel, and Sergey Levine. Soft Actor-Critic: Off-Policy Maximum Entropy Deep Reinforcement Learning with a Stochastic Actor. ICML 2018.
>
> [2] Siliang Zeng, Chenliang Li, Alfredo Garcia, and Mingyi Hong. Maximum-Likelihood Inverse Reinforcement Learning with Finite-Time Guarantees. Decision Awareness in Reinforcement Learning Workshop at the 39th International Conference on Machine Learning (ICML).
>
> [3] Tianwei Ni, Harshit Sikchi, Yufei Wang, Tejus Gupta, Lisa Lee, and Ben Eysenbach. f-IRL: Inverse reinforcement learning via state marginal matching. In Conference on Robot Learning, 2020.
>
> Please let us know if you have any additional concerns or questions.

---

> > ### Comment · Reviewer_K4n8 · 2022-11-19
> > **Acknowleding the response.**
> >
> > Thanks for the response. Many things have been clarified, but my rating does not change based on the response. More comments:
> >
> > 1. Please be aware that PPO is off-policy since it does include an important sampling term. The constraint is unnecessary for an on-policy algorithm.
> >
> > 2. "We note that ERB and EQB yield significant gains over MaxEntIRL in both Figures 1b and 2: In Figure 1b" This is not what I observe. Please double-check. Other reviewers can proofread it.
> >
> > 3. The existence of offline RL clearly weakens the main contributions. Extending offline RL to solve the online problem is natural, although the reverse is problematic.

---

### Official Review · Reviewer_kiih · 2022-10-23

**Confidence:** 4
**Correctness:** 3
**Technical Novelty And Significance:** 2
**Empirical Novelty And Significance:** 2
**Recommendation:** 3

**Clarity, Quality, Novelty And Reproducibility:**

This paper is clearly written and easy to follow, but the idea of incorporating expert guidance in the inner RL training is intuitive.

**Details Of Ethics Concerns:**

No ethics concerns.

**Strength And Weaknesses:**

Strengths:

1. The idea of incorporating expert guidance seem valid to speed up inner RL loop training.
2. Experiment results show improved sample complexity over MaxEntIRL related baselines.

Weaknesses:

1. The approach of incorporating expert demonstrations into inner RL training seems to be intuitive. There is no show of applying expert demonstrations in the target Q value estimation will help recover a valid estimation similar to the ground truth Q value.
2. The experiments do not show performance comparisons with imitation learning baselines such as GAIL.
3. The performance of ERB and EQB is uncertain. Though in most experiments, ERB+EQB works, they fail in the Walker task. The explanation that the surrogate KL objective and return in the task does not correlate is questionable as it does not tell
why f-IRL works in the Walker task but adding ERB+EQB worsens the performance.

**Summary Of The Paper:**

This paper focuses on the high sample and computational complexity problem in IRL which
applies RL in its inner-loop optimization. Therefore, the authors proposed two simple
solutions: 1). Putting expert transitions in the reply buffer of the inner RL algorithm to
familiarize the learner policy with highly rewarded states, i.e., ERB, and 2). Using expert actions
in Q value bootstrapping to improve target Q value estimation, i.e., EQB. Experiment results on
the MuJoCo tasks shows its superior performance when comparing with MaxEntIRL
approaches in improving sample efficiency.

**Summary Of The Review:**

In general, the paper proposes an interesting idea to speed up IRL training in the inner RL training loops with expert guidance. However, the proposed solutions seem incremental by simply incorporating expert demonstrations in reply buffer and target Q value estimation. In addition, it lacks experimental support on its performance over IL baselines, and has performance uncertainties in tasks like Walker.

---

> ### Author Response · Authors · 2022-11-11
> **Response to Reviewer kiih**
>
> We thank the reviewer for all of the helpful comments and suggestions! We address the concerns below:
>  - **“There is no show of applying expert demonstrations in the target Q value estimation will help recover a valid estimation similar to the ground truth Q value.”**: Our EQB method attempts to approximate $Q^*$ (the Q value of the optimal policy) instead of $Q^{\pi}$ (the Q value of the current policy), by maximizing over the learner and expert actions in order to select a more optimal action (See Equation 18) which we believe helps accelerate convergence to the optimal policy and ground truth Q value. (At convergence, taking into account the continuous probability, EQB adds a constant $\log(2)$ term to the $V^{\pi}(s’)$ term in Equation 1)
>  - **“The experiments do not show performance comparisons with imitation learning baselines such as GAIL.”**: While we do not perform explicit performance comparisons with many imitation learning baselines, f-IRL already compares with baselines such as AIRL, f-MAX-RKL, and BC (See the “f-IRL comparisons (imitation learning baselines)” section of our supplementary figures website [here](https://sites.google.com/view/airleb-supp-figures)), demonstrating the superiority of MaxEntIRL and the three variants of f-IRL. Our contribution is an improvement over f-IRL and MaxEntIRL.
>  - **“Though in most experiments, ERB+EQB works, they fail in the Walker task.”**: When we apply ERB+EQB to forward KL (f-IRL), we see performance improvement on the objective being optimized, i.e. the forward KL metric. The mismatch between this metric and performance is certainly interesting, but outside of the scope of this paper. We note that this result is only 1 out of 32 comparisons. We believe that our method has consistently provided significant improvement on most methods and most environments.
>
> Please let us know if you have any additional concerns or questions.

---

### Official Review · Reviewer_vmdh · 2022-10-23

**Confidence:** 3
**Correctness:** 3
**Technical Novelty And Significance:** 2
**Empirical Novelty And Significance:** 2
**Recommendation:** 6

**Clarity, Quality, Novelty And Reproducibility:**

This paper is well-organized and clearly written. I don't have issues reading and understanding the paper. The proposed solution is straightforward, and seems sound. The authors provide a detailed implementation of the simulations. So readers should be able to verify the experiments on their own.

My only concern is with the novelty of this work. As explained in my previous point, most of the existing IRL algorithms do not limit the method used to solve the inner loop RL. There exist efficient off-policy learning methods to solve for an optimal policy with expert data and a fixed, black-box reward function. In this way, the main challenge in this paper is somewhat mitigated.

**Strength And Weaknesses:**

This paper proposes two simple modifications which can significantly accelerate popular IRL algorithms including MaxEntIRL and f-IRL. The authors perform extensive simulations on standard benchmarks like MuJoCo suite of tasks. Simulation results support the authors' conclusion. The simplicity of the solution and the degree of improvement are both appealing.

On the other hand, most IRL algorithms do not restrict the type of data used in the inner loop RL, and can benefit from the expert's demonstrations. Therefore, one could apply off-policy Q-learning (or policy gradient) to obtain an optimal solution for the inner loop RL. For instance (Zhu, Lin, Dai & Zhou, NeurIPS 2020) combined off-policy RL with a popular IRL algorithm called GAIL. In other words, there exist IRL algorithms in the literature that could leverage expert data during the inner loop RL. In this sense, this paper's contribution might be limited.

**Summary Of The Paper:**

This paper proposes a simple set of modifications to two popular inverse-reinforcement learning (IRL) algorithms to improve their computational efficiencies, including MaxEntIRL and f-IRL. Note that a typical IRL algorithm contains two optimization loops: an outer loop that updates a reward function and an inner loop that runs reinforcement learning (RL), usually many steps of policy iteration. Algorithms like MaxEntIRL and f-IRL do not utilize expert data when solving the inner-loop RL problem and instead must act solely based on the black box reward. The authors propose two simple recipes to accelerate MaxEntIRL: (1) placing expert transitions into the replay buffer of the inner RL algorithm and (2) using expert actions in Q value bootstrapping. Simulation results show significant improvement over a MaxEntIRL baseline, which corroborates the authors' claim.

**Summary Of The Review:**

This paper proposes two simple modifications which can significantly accelerate popular IRL algorithms including MaxEntIRL and f-IRL. Simulation results support the authors' conclusion. The simplicity of the solution and the degree of improvement are both appealing. However, most IRL algorithms do not restrict the type of data used in the inner loop RL, and can benefit from the expert's demonstrations using off-policy methods. In this sense, this paper's contribution might be limited. Still, the specific recipes proposed for a popular algorithm like MaxEntIRL could benefit other researchers in the field.

---

> ### Author Response · Authors · 2022-11-11
> **Response to Reviewer vmdh**
>
> Thank you for the encouraging feedback! We address the concerns below:
>  - **“There exist efficient off-policy learning methods to solve for an optimal policy with expert data and a fixed, black-box reward function.”**: We note that the EQB method can be applied in both offline RL and online RL, in the presence of Q updates that use estimated next state Q values as targets to supervise the Q function. The core principle is to build a better approximation to $V^*$ by using multiple potential sources of actions.
>    - In addition, unlike offline RL, the expert policy that collects the expert data in imitation learning is optimal, which differs from the usual assumptions of coverage in offline RL and is likely to lead to suboptimal performance due to covariate shift, as the expert data update distribution may be significantly different from the policy’s actual rollout distribution.
>  - **“For instance (Zhu, Lin, Dai & Zhou, NeurIPS 2020) combined off-policy RL with a popular IRL algorithm called GAIL.”**: OPOLO [1] only utilizes expert data in the RL inner loop by performing BC updates. In comparison, ERB is dynamics aware, which provides built-in resilience to covariate shift.
>
> [1] Zhuangdi Zhu, Kaixiang Lin, Bo Dai, Jiayu Zhou. Off-Policy Imitation Learning from Observations. NeurIPS 2020.
>
> Please let us know if you have any additional concerns or questions.

---

> > ### Comment · Reviewer_vmdh · 2022-11-22
> > **Reply to Authors' Feedback**
> >
> > I thank the author for the response. Some of my concerns have been addressed. Sill, I am not entirely convinced by the novelty of EQB method. The expert's demonstrations could indeed suffer from covariate shifts. However, my point is that effective methods exist in the off-policy literature to address such data shift, including covariate adjustment and importance sampling. From this point of view, this paper's contributions might be somewhat limited.

---

### Official Review · Reviewer_wpP8 · 2022-10-23

**Confidence:** 4
**Correctness:** 3
**Technical Novelty And Significance:** 3
**Empirical Novelty And Significance:** 3
**Recommendation:** 5

**Clarity, Quality, Novelty And Reproducibility:**

- (Related Works) One of the main motivations behind the methods proposed in the paper consists in observing that the IRL process suffers from a high computation and sample demand due to the need for solving an inner RL problem that, in the opinion of the authors, might be more complex than the IRL itself. However, the "Related Works" section completely ignores a class of IRL approaches that do not require the solution of the inner IRL problem, such as [1, 2, 3].

[1] Klein, Edouard, et al. "Inverse reinforcement learning through structured classification." Advances in neural information processing systems 25 (2012).

[2] Pirotta, Matteo, and Marcello Restelli. "Inverse reinforcement learning through policy gradient minimization." Thirtieth AAAI Conference on Artificial Intelligence. 2016.

[3] Ramponi, Giorgia, et al. "Truly batch model-free inverse reinforcement learning about multiple intentions." International Conference on Artificial Intelligence and Statistics. PMLR, 2020.

- (Applicability outside SAC) The authors claim that the two recipes proposed by the paper apply to all actor-critic algorithms. However, the approaches are developed in a formal way for SAC only. While I understand that ERB can be applied whenever there exists a replay buffer, I am having trouble figuring out how to apply EQB outside SAC. Indeed, in Section 5, the authors explain how to define the target for TD learning in the case of SAC only. Can the authors elaborate on how this can be extended for general actor-critic (if possible)?

- (Experiments) As the authors comment, the proposed approaches have some relation with behavioral cloning (BC). Indeed, especially ERB tends to promote the learning of an actor that mimics the expert's actions. In this sense, the experimental evaluation, in my opinion, cannot avoid comparison with BC. Indeed, the remarkable performances shown by the proposed approaches might, in principle, be explained by this effect of inducing the actor towards the expert's actions (which is the principle of BC). The only plot in which BC is shown is Figure 1(b), but, in this case, all versions of MaxEntIRL (with and without the proposed approaches) behave in an almost similar way.

Furthermore, I couldn't find the number of runs employed to generate the plots as well as the meaning of the shaded areas (are they confidence intervals? standard deviation?)


**Minor Issues**
- The notation $\tau \sim \pi$ is not explained. I understand that it means that trajectory $\tau$ is sampled from the trajectory distribution induced by policy $\pi$, but this should made clear.
- Missing punctuation in most of the full-line equations.

**Details Of Ethics Concerns:**

None.

**Strength And Weaknesses:**

**Strengths**
- The experimental evaluation shows the two approaches proposed in the paper are able to significantly outperform the plain MaxEntIRL in both the toy example and the Mujoco environments.
- The approach seems to have the potential to be applied even for other learning algorithms besides SAC.

**Weaknessees**
- The approach is developed in detail assuming that the inner learning algorithm is SAC. It is not straightforward to generalize some aspects to general actor-critic.
- It is not clear how these remarkable results compare with standard behavioral cloning.
- The proposed approaches can be considered incremental w.r.t. the IRL algorithms to which they are applied. Indeed, although reasonable and easy to understand, no guarantee from a theoretical perspective is provided.

**Summary Of The Paper:**

The paper proposes a method to accelerate the inverse reinforcement learning (IRL) process, by injecting knowledge about the expert's policy in two different ways. The first idea consists in introducing transitions from the expert's demonstrations in the replay buffer (expert replay bootstrapping, ERB). The second idea consists in employing the expert's actions as next actions for defining the target in the learning of the critic (expert Q bootstrapping, EQB). Both ideas are developed formally and tested. The evaluation is performed first on a toy example to illustrate the functioning of the methods and, then, on four Mujoco environments in combination with MaxEntIRL.

**Summary Of The Review:**

Overall, the paper proposes reasonable modifications to IRL algorithms coupled with actor-critic methods, in order to speed up the IRL process. Although the experimental evaluation succeeds in showing the advantages compared with the plain IRL, there is no comparison with BC. Moreover, the proposed modifications are, in my opinion, incremental. Therefore, at the moment I opt for a borderline negative evaluation.

---

> ### Author Response · Authors · 2022-11-11
> **Response to Reviewer wpP8**
>
> We really appreciate your valuable time and feedback in this review! We address your concerns below:
>
>  - **“However, the "Related Works" section completely ignores a class of IRL approaches that do not require the solution of the inner IRL problem”**: We thank the reviewer for the suggestion and will add discussion regarding the IRL approaches that do not require a solution to the inner RL loop problem.
>  - **“Can the authors elaborate on how this can be extended for general actor-critic (if possible)?”**: Our methods are generalizable to any actor-critic methods where the Q updates bootstrap using the estimated Q value at the next state as a target. In these cases, we simply need to replace this next-state target with the EQB target as defined in Equation 10, which uses both expert and learner actions on updates on expert transitions.
>    - For example, EQB could be extended to non-maximum entropy actor-critic methods (e.g. DDPG, TD3, DQN) by simply taking the maximum between the Q values for the expert and learner actions.
>  - **“In this sense, the experimental evaluation, in my opinion, cannot avoid comparison with BC.”**: Though we do not explicitly compare against BC in our paper, f-IRL[1] compares against BC on all four environments and shows the superiority of MaxEntIRL and f-IRL over BC, which often does not achieve expert performance levels. In addition, our BC results indicate that stochastic BC is unable to recover to expert-level return on 3 of the 4 environments (See the “BC results” and “f-IRL comparisons (imitation learning baselines)” sections on our supplementary figures website [here](https://sites.google.com/view/airleb-supp-figures) for the f-IRL paper results and our results) We suspect this is due to covariate shift, as the expert distribution on which updates are done may be different from the policy rollout distribution. Our methods achieve the best of both worlds - significantly accelerating learning using expert data while being dynamics-aware and preventing covariate shift.
>  - **“The only plot in which BC is shown is Figure 1(b), but, in this case, all versions of MaxEntIRL (with and without the proposed approaches) behave in an almost similar way.”**: Figure 1b shows that MaxEntIRL+ERB and EQB are significantly faster at recovering large levels of expert performance at the beginning of training.
>
> We thank the reviewer for pointing out issues with the draft and will update it accordingly:
>  - **“Furthermore, I couldn't find the number of runs employed to generate the plots as well as the meaning of the shaded areas (are they confidence intervals? standard deviation?)”**: As mentioned in the Experiments section, our results are averaged over 5 seeds. The shaded areas are standard deviations. We will make that clear in the captions.
>  - **“The notation τ∼π is not explained.”**: We will add clarification regarding the notation $\tau \sim \pi$ in the Background section.
>  - **“Missing punctuation in most of the full-line equations.”**: We will add punctuation to our equations.
>
> [1] Tianwei Ni, Harshit Sikchi, Yufei Wang, Tejus Gupta, Lisa Lee, and Ben Eysenbach. f-IRL: Inverse reinforcement learning via state marginal matching. In Conference on Robot Learning, 2020.
>
> Please let us know if you have any additional concerns or questions.

---

> > ### Comment · Reviewer_wpP8 · 2022-11-19
> > **Reply to Authors' Feedback**
> >
> > I thank the authors for the provided feedback. I have read it together with the other reviews. Most of my concerns have been clarified. Still, I think that the paper's contribution is quite incremental and there is room for improvement in the organization (especially the explicit presentation of the approach beyond SAC). I will consider adjusting my score after the discussion with the other reviewers.

---

### Author Response · Authors · 2022-11-11
**Supplementary Figures**

We thank all the reviewers for their detailed and thoughtful feedback. We have published an additional website with supplementary results for Behavior Cloning, IQ-Learn, and other imitation learning baselines to address questions and concerns. You can find it [here](https://sites.google.com/view/airleb-supp-figures). Thank you!

---

### Decision · Program_Chairs · 2023-01-20

**Decision:**

Reject

**Justification For Why Not Higher Score:**

The paper has limited novelty and more comparisons with related works are needed.
These modifications require time and another round of reviews.

**Justification For Why Not Lower Score:**

N/A

**Metareview: Summary, Strengths And Weaknesses:**

The paper proposes two ways to accelerate the inner loop of Inverse Reinforcement Learning algorithms by leveraging the knowledge of the expert's policy. The proposed solution is tested both on a toy problem and four Mujoco environments.
Although the idea of speeding up the inner loop with knowledge of the expert policy is sensible and effective, the reviewers have raised several issues about the novelty and the significance of the proposed approach.
Furthermore, they suggest adding comparisons with important baselines to best assess the advantage of the proposed approach.
After reading each others' reviews and the authors' feedback, part of the reviewers' concerns have been solved, but the reviewers agree that further work is needed to make this paper ready for publication.
We encourage the authors to consider the suggestions provided in the reviews while preparing a new version of their paper.